# Preparation of Nickel Oxide Nanoflakes for Carrier Extraction and Transport in Perovskite Solar Cells

**DOI:** 10.3390/nano12193336

**Published:** 2022-09-25

**Authors:** Chih-Yu Chang, You-Wei Wu, Sheng-Hsiung Yang, Ibrahim Abdulhalim

**Affiliations:** 1Institute of Lighting and Energy Photonics, College of Photonics, National Yang Ming Chiao Tung University, No. 301, Section 2, Gaofa 3rd Road, Guiren District, Tainan 71150, Taiwan; 2Department of Electro-Optics and Photonics Engineering and the Ilse Katz Institute for Nanoscale Science and Technology, School of Electrical and Computer Engineering, Ben-Gurion University of the Negev, Beer-Sheva 8410501, Israel

**Keywords:** nickel oxide, hole transport layer, nanoflakes, carrier extraction, inverted perovskite solar cells

## Abstract

Hole transport layers (HTLs) with high conductivity, charge extraction ability, and carrier transport capability are highly important for fabricating perovskite solar cells (PSCs) with high power conversion efficiency and device stability. Low interfacial recombination between the HTL and perovskite absorber is also crucial to the device performance of PSCs. In this work, we developed a three-stage method to prepare NiO_x_ nanoflakes as the HTL in the inverted PSCs. Due to the addition of the nanoflake layer, the deposited perovskite films with larger grain sizes and fewer boundaries were obtained, implying higher photogenerated current and fill factors in our PSCs. Meanwhile, the downshifted valence band of the NiO_x_ HTL improved hole extraction from the perovskite absorber and open-circuit voltages of PSCs. The optimized device based on the NiO_x_ nanoflakes showed the highest efficiency of 14.21% and a small hysteresis, which outperformed the NiO_x_ thin film as the HTL. Furthermore, the device maintained 83% of its initial efficiency after 60 days of storage. Our results suggest that NiO_x_ nanoflakes provide great potential for constructing PSCs with high efficiency and long-term stability.

## 1. Introduction

Organometallic halide perovskites have drawn a lot of attention owing to their high molar extinction coefficient, broad light absorption range, long diffusion length, and tunable bandgap [1,2,3,4,5]. Furthermore, the cost-effective solution processability makes perovskite solar cells (PSCs) a good candidate for next-generation solar cells [6,7]. Developments in perovskite absorbers and carrier transport materials, as well as the progress in fabrication techniques, gradually boost the performance of PSCs. Recently PSCs have been recognized worldwide for their record power conversion efficiency (PCE) of over 25% [8].

The device architecture of PSCs is usually categorized into two types, i.e., traditional planar (n-i-p) and inverted (p-i-n) structures [9,10]. So far, most of the record-breaking PSCs have been constructed with the traditional n-i-p structure using compact and mesoporous TiO_2_ layers as the electron transport layer (ETL), which provide good electron extraction and transport capabilities [11,12,13]. The existence of the mesoporous layer is considered to enlarge the interfacial electron/hole separation zone, thereby improving the device’s efficiency. However, the severe current density–voltage (J–V) hysteresis usually occurs for PSCs with the n-i-p structure, which limits the power output of the device [14,15]. On the contrary, PSCs with the p-i-n structure are found to possess little or even negligible hysteresis due to the enhanced hole extraction and transport in this kind of device structure [16]. Furthermore, it is reported that holes in perovskites have a shorter diffusion length than electrons, which can be compensated in the p-i-n device structure [17,18]. Besides, the inverted structure offers the advantage of low-temperature processability, thereby reducing manufacturing cost and particularly bringing compatibility with flexible substrates to expand the diversity of photovoltaic cells. Adopting a suitable hole transport layer (HTL) is extremely important for p-i-n PSCs, which can decrease the loss of photogenerated currents and prevent charge recombination at the HTL/perovskite interface.

Conventional organic polymers such as poly(3,4-ethylenedioxythiophene): polystyrenesulfonate (PEDOT:PSS) [19,20,21], poly [bis (4-phenyl) (2,4,6-trimethylphenyl) amine] (PTAA) [22], and poly [(9,9-dioctylfluorenyl-2,7-diyl)-co-(4,4′-(N-(4-sec-butylphenyl) diphenylamine)] (TFB) [23], have been utilized as the HTL for manufacturing inverted PSCs. Despite the high photovoltaic performance of PSCs, the relatively high price and instability of organic polymers under continuous solar irradiation hinder the commercialization of PSCs using organic HTLs. In contrast, inorganic HTLs, including NiO_x_ [24,25,26], CuO_x_ [27], and CuSCN [28], are promising alternatives to construct PSCs with high efficiency and environmental stability. Among them, naturally abundant NiO_x_ has been largely utilized as the HTL in inverted PSCs. The replacement of organic HTLs with inorganic ones can bring better stability and minimize charge loss. To further facilitate the development of NiO_x_-based devices, superior hole mobility, as well as carrier extraction ability, is crucial for NiO_x_ HTLs. Based on a literature survey, we realize that most of the recent works have focused on metal ion-doped NiO_x_ with improved conductivity for better device performance [29,30,31]; however, ionic doping of the NiO_x_ HTLs is considered to cause lattice disorder and defects that may result in decreased carrier mobility and parasite recombination [32]. Therefore, it is of great scientific interest to develop high-quality NiO_x_ HTLs for the fabrication of PSCs without ionic doping. Different NiO_x_ nanostructures have been proposed and utilized as the HTL in inverted PSCs. Yin et al. demonstrated NiO_x_ nanoparticles by a solvothermal method [33], which was spin-cast into thin film from a colloid precursor solution and annealed at 300 °C to serve as an efficient HTL. Hysteresis-less PSCs based on the NiO_x_ HTL showed a high PCE of 16.68% and a steady-state efficiency of 16.49%. Song, Xiong, and their co-workers proposed the NiO nanotube nanoforest as an efficient hole extraction layer [34]. The mesoporous architecture supplies highly conductive pathways for effective hole extraction and inhibits charge recombination at the interface. The optimized device exhibited a high efficiency of 18.77% and retained 79% of its initial PCE after a three-step aging period of 600 h. In 2020, they reported a NiO nanowall film via a one-step hydrothermal method and thermal annealing at 350 °C [35]. The prepared NiO nanowall film exhibited a dense and mesoporous structure which ameliorated the contact at the perovskite/NiO interface and passivated interfacial defects. An optimal PCE of 17.8% was obtained, and it was further augmented to 19.16% by introducing a diethanolamine interlayer on the NiO nanowall film. Our group reported the nanoporous NiO_x_ layer via the chemical bath deposition in 2021 [36]. The sponge-like nanostructure helps to grow high-quality perovskite films and improves carrier extraction from the perovskite absorber. The device showed a moderate PCE of 13.43% and maintained 80% of its initial PCE after 50 days of storage. In addition to the above literature, Hong and his co-workers also reported inverted PSCs based on compact or nanoporous NiO_x_ with a PCE of 15.35–19.1% [37,38]. It is worth developing NiO_x_ HTL with different mesoporous nanostructures for applications in PSCs to pursue high efficiency and environmental stability.

In this work, we demonstrate NiO_x_ nanoflakes by the three-stage method using ZnO nanorods as a sacrificial template. We have a lot of research experience in synthesizing ZnO nanorods for the construction of photovoltaic and light-emitting devices. It is well known that ZnO can be etched by strong bases such as sodium hydroxide (NaOH) or potassium hydroxide. We presume that mesoporous NiO_x_ can be formed on top of ZnO nanorods; by chemical etching with a strong base, the uncovered ZnO could be removed, and new NiO_x_ nanostructures might be obtained. Therefore, we performed the experiment and obtained NiO_x_ nanoflakes with different dimensions successfully, which was confirmed through SEM and AFM observations. The schematic illustration of the preparation of NiO_x_ nanoflakes is depicted in Figure 1. In the step (1), ZnO nanorods were grown on the FTO substrate by the hydrothermal method. In the step (2), a mesoporous NiO_x_ layer was established on top of ZnO nanorods by the chemical bath deposition. In the step (3), ZnO was removed by chemical etching and consequently NiO_x_ nanoflakes were formed. The NiO_x_ nanoflakes were utilized as the efficient HTL for fabricating inverted PSCs. The enhanced hole mobility and electrical conductivity of the obtained NiO_x_ nanoflakes facilitate interfacial charge transfer and reduce carrier recombination compared with the NiO_x_ thin film. The sponge-like structure is favorable for hole extraction and charge dissociation. In addition, an ultrathin PTAA film was inserted between NiO_x_ nanoflakes HTL and the perovskite film to effectively passivate interfacial defects and increase device performance. The optimized inverted PSC based on the NiO_x_ nanoflakes HTL exhibited superior long-term stability and possessed 83% of its initial efficiency over 60 days of storage.

## 2. Experimental Section

### 2.1. Materials

The patterned fluorine-doped tin oxide (FTO, 7 Ω/square)-coated substrates were provided by LiveStrong Optoelectronics Technology Co., Ltd. (Kaohsiung, Taiwan). Zinc acetate dihydrate (Zn(OAc)_2_•2H_2_O, purity 98–101%), nickel acetate tetrahydrate (Ni(OAc)_2_•4H_2_O, purity 98+%), and nickel chloride (NiCl_2_, purity 99%) were bought from Alfa Aesar (Ward Hill, MA, USA). Zinc sulfate heptahydrate (ZnSO_4_•7H_2_O, purity 99.5%) and potassium persulfate (K_2_S_2_O_8_, purity 98%) were purchased from Showa (Tokyo, Japan). Perovskite precursors, including lead bromide (PbBr_2_, purity 98+%) and cesium iodide (CsI, purity 99.9%), were brought from Alfa Aesar. Lead iodide (PbI_2_, purity 99.9985%), ethanolamine (purity 99%), and tetrabutylammonium tetrafluoroborate (TBABF_4_, purity 98%) were purchased from Acros (Geel, Belgium). Methylammonium bromide (MABr, purity 99.5%) and formamidinium iodide (FAI, purity 98%) were brought from STAREK Scientific Co., Ltd. (Kaohsiung, Taiwan) and Lumtec (Hsinchu, Taiwan), respectively. Poly [bis (4-phenyl) (2,4,6- trimethylphenyl) amine] (PTAA, molecular weight 6000–15,000 g/mol), polyethyleneimine (PEI, molecular weight 25,000 g/mol), and [6,6]-phenyl-C_61_-butyric acid methyl ester (PC_61_BM, purity 99%) were bought from Xi’an Polymer Light Technology Corp. (Shaanxi, China), Sigma-Aldrich (St. Louis, MO, USA), and Solenne B. V. (Groningen, The Netherlands), respectively. Other organic solvents were received from Acros or Alfa Aesar and used without further purification.

### 2.2. Preparation of NiO_x_ Thin Films and Nanoflakes

To prepare NiO_x_ thin films, a precursor solution consisting of Ni(OAc)_2_•4H_2_O (0.124 mg) and ethanolamine (30 μL) in 5 mL of isopropyl alcohol (IPA) was heated at 70 °C with stirring in a sealed glass vial overnight. The precursor solution was then spin-coated on the FTO substrate, followed by calcination at 450 °C to obtain the NiO_x_ thin film.

The growth of ZnO nanorods on the FTO substrate was referred to in our previous report [39]. Afterward, mesoporous NiO_x_ was formed on ZnO nanorods via the chemical bath deposition and chemical etching. The detailed preparation process of mesoporous NiO_x_ is described as follows. The substrate was immersed in an aqueous solution containing NiCl_2_ (777.4 mg), K_2_S_2_O_8_ (162.2 mg), and deionized (DI) water (150 mL), followed by transferring the solution into an oven and heating at 65 °C for 20 min for NiO_x_ growth. The NiO_x_-covered substrate was then taken out and immersed in a 0.5 M NaOH aqueous solution for 30 min to remove ZnO nanorods. Finally, the substrate was rinsed with DI water, dried with nitrogen flow, and calcined at 350 °C in a high-temperature oven for 1 h to obtain NiO_x_ nanoflakes.

### 2.3. Device Fabrication

FTO substrates were sequentially cleaned in detergent, DI water, acetone, and IPA under ultra-sonication for 15 min each, followed by UV-ozone treatment for 25 min before the growth of ZnO nanorods or the deposition of the NiO_x_ thin film. A thin PTAA layer was deposited on top of the NiO_x_ thin film or NiO_x_ nanoflakes by spin-coating at 5000 rpm for 60 s from its solution (0.5 mg/mL in toluene) and dried at 100 °C for 10 min. The perovskite layer was then deposited on top of the NiO_x_ thin film/PTAA or NiO_x_ nanoflakes/PTAA, followed by the deposition of the PCBM+TBABF_4_ layer. The details for the preparation and deposition of the perovskite and PCBM+TBABF_4_ layers can be referred to in our previous report [36]. Afterward, 0.1 wt% of PEI in IPA was spin-coated on the PCBM+TBABF_4_ layer at 5000 rpm for 30 s. Finally, Ag electrodes (100 nm) were thermally evaporated at a high vacuum pressure of 6 × 10^−6^ torr. The effective area of each device is 4.5 mm^2^.

### 2.4. Characterization Methods

The cross-sectional and top-view scanning electron microscopy (SEM) images of the NiO_x_ thin film and nanoflakes were obtained on an ultrahigh-resolution ZEISS AURIGA Crossbeam scanning electron microscope (Oberkochen, Germany). The energy dispersive X-ray spectroscopy (EDS, Bruker Quantax, Billerica, MA, USA) spectra of samples were obtained on the same SEM instrument. The morphology and surface roughness of the NiO_x_ thin film and nanoflakes were measured with the tapping mode on a Bruker Innova atomic force microscope (AFM, Billerica, MA, USA). The transmission and absorption spectra of the two NiO_x_ samples were obtained from a Princeton Instruments Acton 2150 spectrophotometer (Acton, MA, USA). A xenon lamp (ABET Technologies LS 150, Milford, CT, USA) was used as the excitation source. The ultraviolet photoelectron spectroscopy (UPS) experiments for the NiO_x_ thin film and nanoflakes were obtained from a Thermo VG-Scientific/Sigma Probe instrument (Waltham, MA, USA), using a He I discharge lamp (Berlin, Germany) as the excitation source (hν = 21.22 eV). X-ray diffraction (XRD) patterns and crystallinity of NiO_x_ and perovskites were acquired on a Rigaku D/MAX2500 X-ray diffractometer (Tokyo, Japan). Elemental composition analysis of samples was performed on a Thermo K-Alpha X-ray photoelectron spectrometer (XPS, Waltham, MA, USA). The photoluminescence (PL) emission spectra of the perovskites on the FTO, PTAA, or different NiO_x_/PTAA layers were examined by a Princeton Instruments Acton 2150 spectrophotometer. A KIMMON KOHA He-Cd laser (Tokyo, Japan) with double excitation wavelengths (325/442 nm) was chosen as the light source. The time-resolved PL (TR-PL) signals were acquired on a time-correlated single-photon counting module (PicoQuant MultiHarp 150 4N, Berlin, Germany), combined with a photomultiplier tube through an Andor Kymera 328i spectrometer (Belfast, Northern Ireland, United Kindom. The excitation source for the TR-PL decay experiment is a pulsed laser (Omicron, Rodgau, Germany) at 473 nm. The J–V curves of the fabricated PSCs were acquired using a Keithley 2400 source measuring unit (Beaverton, Oregon, USA) with a scan rate of 30 mV/s under AM 1.5 G simulated sunlight exposure (Yamashita Denso YSS-150A, Tokyo, Japan, using a 1000 W xenon short arc lamp). The devices were measured at 100 mW/cm^2^ intensity under ambient air. The external quantum efficiency (EQE), as well as integrated current density, was obtained on an assembled system comprising a Keithley 2400 source measuring unit, a xenon lamp (ABET Technologies LS 150), and a monochromator (Prince Instruments Acton 2150).

## 3. Results and Discussion

### 3.1. Characterization of NiO_x_ Thin Film and Nanoflakes

In this study, NiO_x_ nanoflakes were prepared by the three-stage method, as revealed in Figure 1. In the first stage, ZnO nanorods were grown on the FTO substrate by the hydrothermal method. In the second stage, the substrate was immersed in the Ni precursor solution to form a mesoporous NiO_x_ layer on top of ZnO nanorods. In the third stage, the NiO_x_ layer-covered substrate was immersed in a NaOH aqueous solution to remove ZnO by chemical etching, followed by high-temperature calcination to obtain NiO_x_ nanoflakes. The morphological changes at different stages were investigated by SEM experiments, and corresponding SEM images are provided in Appendix A in the Supporting Information. Short and compact ZnO nanorods were formed on the FTO surface via the hydrothermal method. Mesoporous NiO_x_ was then formed on top of ZnO nanorods. After ZnO etching, a different NiO_x_ nanostructure with less dense porosity was obtained.

Figure 2 shows the top-view and cross-sectional SEM micrographs of the NiO_x_ thin film and nanoflakes. The NiO_x_ thin film was prepared according to the previous report for comparison [40]. A thin and dense NiO_x_ layer was deposited on the surface of FTO, and low-lying FTO grains are clearly seen, as shown in Figure 2a. The thickness of the NiO_x_ layer is estimated to be 30 nm in Figure 2c. As for the NiO_x_ nanoflakes, the sponge-like nanostructure is observed, and the height of interconnecting networks is about 58 nm, as revealed in Figure 2b,d, respectively. We expect that NiO_x_ nanoflakes with large surface areas are profitable for the formation of high-quality perovskite layers. To verify the existence of Zn, we conducted an EDS analysis of the prepared NiO_x_ nanoflakes, and the corresponding EDS spectrum is provided in Appendix A. It is seen that a small amount of Zn element exists in the NiO_x_ nanoflakes. Appendix A presents the topographic AFM images of the NiO_x_ thin film and nanoflakes, revealing similar morphologies to the top-view SEM images. The average surface roughness (*R*_a_) of the NiO_x_ thin film and nanoflakes is estimated to be 16.2 and 27.5 nm, respectively.

Figure 3 shows the transmission and absorption spectra of the NiO_x_ film and nanoflakes from 300 to 750 nm. In Figure 3a, the prepared NiO_x_ film has a moderate transmittance of 60–80% from 350 to 460 nm and a higher transmittance of 80–90% from 460 to 750 nm. Moreover, the NiO_x_ nanoflakes layer reveals a similar spectral line but lower transmittance than the NiO_x_ film. The lower transmittance of the NiO_x_ nanoflakes can be ascribed to the thicker thickness of the nanoflakes layer that has been confirmed by SEM. Nevertheless, our NiO_x_ nanoflakes with moderate transmittance in the visible range can still be utilized as the HTL for incident light to enter devices and be absorbed by the perovskite layer. In Figure 3b, both the NiO_x_ film and nanoflakes possess similar absorption spectra, while the latter has slightly higher absorbance due to the thicker layer. The optical bandgaps (*E*_g_) of the NiO_x_ thin film and nanoflakes are approximated by their absorption edges at around 350 nm to be 3.61 and 3.57 eV, respectively, which are in accordance with our previous reports [36,40].

The UPS spectra of the NiO_x_ thin film and nanoflakes displayed in Figure 4 to examine whether the energy levels of NiO_x_ were altered due to different nanostructures. The high binding energy cutoff of the NiO_x_ thin film and nanoflakes in Figure 4a was determined at 16.25 and 16.07 eV, respectively. The Fermi levels (E_F_) of the NiO_x_ thin film and nanoflakes were then calculated to be −4.97 and −5.15 eV, respectively, by subtracting the high binding energy cutoff from the He I photon energy (21.22 eV) [41]. In Figure 4b, the low binding energy cutoffs were found at 0.21 and 0.12 eV for the NiO_x_ thin film and nanoflakes, respectively. The valence band (VB) levels of the NiO_x_ thin film and nanoflakes were then calculated to be −5.18 and −5.27 eV, respectively, since the low binding energy cutoff indicates the energy difference between the E_F_ and the VB. Therefore, the nanoflakes structure resulted in a favorable downward shift of the VB level, which is beneficial to hole extraction from the perovskite layer to the NiO_x_ HTL owing to better band alignment [40]. The conduction band (CB) levels of the NiO_x_ thin film and nanoflakes were determined to be −1.57 and −1.7 eV, respectively, from their E_g_ values.

Figure 5 shows the XRD patterns of the NiO_x_ thin film and nanoflakes on the FTO substrates. It is noted that seven labelled (*) peaks are originated from FTO. The NiO_x_ thin film has three diffraction peaks at 2θ = 38.9, 43.3, and 62.8°, which correspond to the (111), (200), and (220) planes, respectively. After analyzing the XRD patterns, the formed NiO_x_ thin film is assigned to the cubic phase [42]. Moreover, the NiO_x_ nanoflakes also possess three diffraction peaks at similar positions of 2θ = 38.9, 43.1, and 62.9°, indicating that both NiO_x_ thin film and nanoflakes have the same crystalline structure. To further investigate Ni^3+^ and Ni^2+^ species in the NiO_x_ thin film and nanoflakes, XPS experiments were carried out, and the Ni *2p_3/2_* and O *1s* signals are revealed in Figure 6. According to the previous literature [40,43], the multicomponent band of the Ni element was deconvoluted into four different states at 854.1 (Ni^2+^), 855.8 (Ni^3+^), 860.8 (Ni^2+^ satellite), and 863.7 eV (Ni^3+^ satellite). The Ni^3+^/Ni^2+^ ratios were then determined to be 1.19 and 1.49 for the NiO_x_ thin film and nanoflakes from Figure 6a,b, respectively. The significantly increased Ni^3+^/Ni^2+^ ratio explains the fact that the NiO_x_ nanoflakes layer has a higher hole transport ability than the NiO_x_ thin film. It is certain that higher Ni^3+^/Ni^2+^ ratio means a higher number of holes in the NiO_x_ nanoflakes than in thin film. The reason for the higher hole transport ability in the NiO_x_ nanoflakes is described as follows. The higher Ni^3+^ concentration in the NiO_x_ nanoflakes implies the existence of interstitial oxygen and Ni vacancies in the crystalline structure. Compared with the Ni^2+^, the higher valence Ni^3+^ means a lack of one electron that produces a hole carrier. These hole carriers can move through the crystal by hopping from the Ni^3+^ site to another one or by moving in a narrow polaron band, indicative of enhanced hole transport ability [40]. Figure 6c,d demonstrate the O *1s* spectra of the NiO_x_ thin film and nanoflakes, respectively, which are deconvoluted into two states at 529.3 (from lattice oxygen) and 531.3 eV (from surface O–H groups) [43].

### 3.2. Characterization of Perovskite Layers on NiO_x_

Figure 7a,b show the cross-sectional SEM micrographs of the whole devices using the NiO_x_ thin film and nanoflakes as the HTL, respectively. The perovskite layer has a thickness of 550 nm on both NiO_x_. Moreover, the perovskite nanocrystals with larger grain sizes and fewer grain boundaries were obtained on the NiO_x_ nanoflakes, meaning that fewer defects were produced as compared with the NiO_x_ thin film. Figure 7c,d reveal the top-view SEM micrographs of the perovskite on the NiO_x_ film/PTAA and NiO_x_ nanoflakes/PTAA, respectively. It is noted that a thin PTAA layer was inserted between NiO_x_ and the perovskite layer for the two samples. The grain size of the perovskite deposited on the NiO_x_ thin film/PTAA is observed to be 200–300 nm; besides, a larger grain size of 400–500 nm is obtained when deposited on the NiO_x_ nanoflakes/PTAA. Apart from SEM observation, the AFM technique was also applied to investigate topologies of the perovskite on the NiO_x_ film/PTAA and NiO_x_ nanoflakes/PTAA, as revealed in Appendix A. It is seen that the grains of the perovskite on the NiO_x_ nanoflakes/PTAA look larger and more uniform than those on the NiO_x_ thin film/PTAA, which is consistent with SEM observation. As mentioned in the previous part, NiO_x_ nanoflakes with a porous nanostructure are expected to serve as a template for high-quality perovskite deposition, which is confirmed by SEM observation. Perovskite films with larger grain sizes and fewer boundaries are useful for decreasing charge recombination and promoting carrier extraction from the perovskite absorber to charge transport layers. Figure 8 shows the XRD patterns of the perovskite Cs_0.05_FA_0.81_MA_0.14_Pb(Br_0.15_I_0.85_)_3_ on the NiO_x_ thin film/PTAA and NiO_x_ nanoflakes/PTAA. Several diffraction peaks are located at 2θ = 14.02, 19.89, 24.43, 28.29, 31.74, 34.90, 40.56, and 43.15°, corresponding to the (001), (011), (111), (002), (012), (112), (022), and (003) planes. The positions of these diffraction signals are in line with the previous literature [36,44]. We conclude that the formation of perovskite crystals was not affected by different low-lying NiO_x_ nanostructures.

The absorption spectra of the perovskite on PTAA, NiO_x_ thin film/PTAA, and NiO_x_ nanoflakes/PTAA are depicted in Appendix A, which look similar from 550 to 800 nm. The perovskite deposited on the NiO_x_ nanoflakes/PTAA has the strongest absorbance among the three samples, which is favorable for improving J_SC_ and the performance of PSCs. Figure 9a shows the PL emission spectra of the perovskite on the FTO, PTAA, NiO_x_ film/PTAA, and NiO_x_ nanoflakes/PTAA. The PL emission of the four perovskite films on different substrates is centered at 770 nm, which is in accordance with the previous report [45]. It is seen that the perovskite exhibits the highest PL intensity on the FTO substrate. With the insertion of PTAA, the PL intensity of the perovskite is slightly lowered. Significant PL quenching is observed when depositing the perovskite on the NiO_x_ film/PTAA, indicative of carrier extraction at the perovskite and NiO_x_/PTAA interface. Moreover, the perovskite on the NiO_x_ nanoflakes/PTAA shows even stronger PL quenching and the lowest PL intensity, implying more efficient carrier extraction and reduced recombination that is helpful to improve the J_SC_ of PSCs. To examine charge transfer at the interface between the perovskite and NiO_x_/PTAA, the TR-PL measurement was conducted, and corresponding TR-PL decay curves of the perovskite on the FTO, PTAA, NiO_x_ films/PTAA, and NiO_x_ nanoflakes/PTAA are depicted in Figure 9b. It is seen that the perovskite on the NiO_x_ nanoflakes/PTAA has the fastest PL decay among the four samples, revealing low recombination and efficient charge separation [46,47]. The TR-PL decay curves agree well with a biexponential decay fitting and correspond-ing fast-decay τ_1_, slow-decay τ_2_, and average lifetime (τ_avg_) are listed in Appendix A. The perovskite on the FTO, PTAA, NiO_x_ thin film/PTAA, and NiO_x_ nanoflakes/PTAA has an average lifetime (τ_avg_) of 139.5, 91, 72.3, and 53.3 ns, respectively. The TR-PL results are consistent with PL observation, indicating a faster charge transfer and more efficient carrier extraction from the perovskite active layer by the NiO_x_ nanoflakes/PTAA compared with the NiO_x_ thin film/PTAA.

### 3.3. Device Evaluation of PSCs

The p-i-n devices with the structure of FTO/NiO_x_ thin film or nanoflakes/PTAA/Cs_0.05_(MA_0.85_FA_0.15_)_0.95_Pb(Br_0.15_I_0.85_)_3_/PCBM+TBABF_4_/PEI/Ag were constructed and evaluated. The active area of each device is 4.5 mm^2^. In our previous reports, the incorporation of TBABF_4_ in PCBM could improve electron transport ability at the perovskite/PCBM interface [40,48]. In addition, PEI was employed as an interfacial layer to improve electron extraction between PCBM and metal electrodes [49]. Figure 10a shows the energy level diagram of the whole PSC. The VB and CB levels of the NiO_x_ thin film and nanoflakes are estimated and listed in the Section 3.1, while the energy levels of the remaining layers are referred to in the previous reports [48,49,50]. In our designed device, electrons can be migrated smoothly from the perovskite absorber to the Ag electrode through PCBM+TBABF_4_/PEI. At the same time, dissociated holes are transported from the perovskite absorbing layer to the FTO electrode through NiO_x_ stepwise. Furthermore, a downshifted VB level of the NiO_x_ nanoflakes HTL will lead to higher V_OC_ values. The J–V curves of PSCs in the forward and reverse directions under AM 1.5 G illumination are displayed in Figure 10b, and the several photovoltaic parameters including V_OC_, J_SC_, FF, PCE, series resistance (R_S_), and shunt resistance (R_Sh_) are listed in Table 1. The champion PSC using the NiO_x_ nanoflakes HTL has a V_OC_ of 0.99 V in the reverse scan, a J_SC_ of 20.5 mA/cm^2^, a FF of 70%, and a PCE of 14.21%, which is significantly better than the device based on the NiO_x_ thin film (J_SC_ = 19.1 mA/cm^2^, V_OC_ = 0.94 V, FF = 66%, and PCE = 11.88%). The performance enhancement is mainly attributed to the augmented J_SC_ and V_OC_. For comparison, the device using bare PTAA as the HTL was also fabricated and evaluated. The corresponding J–V curves in both scanning directions are provided in Appendix A, revealing a J_SC_ of 16.14 mA/cm^2^, a V_OC_ of 0.95 V, a FF of 60%, and a PCE of 9.1% in the reverse scan. The PCE of the device based on bare PTAA is even lower in the forward scan (8.6%). We conclude that the incorporation of NiO_x_ HTLs is effective in enhancing the device performance of PSCs. In the early stage of this study, we utilized a sole NiO_x_ thin film or nanoflakes as the HTL for fabricating PSCs and received lower device performance (PCE = 10.41% for NiO_x_ thin film and 12.44% for NiO_x_ nanoflakes). The J–V characteristics of devices without PTAA are provided in Appendix A. According to the previous literature [51], we realized that a thin layer of PTAA can be inserted between NiO_x_ and the perovskite layer to improve carrier transport ability and photovoltaic properties. Therefore, PTAA was incorporated into all devices, and higher device performance was received. Moreover, PSCs using the NiO_x_ nanoflakes HTL exhibited smaller hysteresis than those based on the NiO_x_ thin film. It is reported that the hysteresis effect originates from charge accumulation at the interface, high defect density, and/or unbalanced charge transport in PSCs [52]. The hysteresis index (HI) is proposed as HI = (PCE_reverse_—PCE_forward_)/PCE_reverse_ [53], which is estimated to be 0.077 and 0.017 for PSCs based on the NiO_x_ thin film and NiO_x_ nanoflakes, respectively. The hysteresis phenomenon is eliminated by reducing surface defects of perovskites, which has been discussed in the previous section. Apart from reduced defects, more balanced charge transport in the NiO_x_ nanoflakes-based device is also responsible for a smaller hysteresis effect. To investigate charge transport behaviors, additional devices, FTO/NiO_x_ thin film or nanoflakes/Ag and FTO/PCBM+TBABF_4_/PEI/Ag, were fabricated and compared. The corresponding current-voltage characteristics are depicted in Appendix A. It is seen that the hole-only device FTO/ NiO_x_ nanoflakes/Ag shows very close conductivity to the electron-only device FTO/PCBM+TBABF_4_/PEI/Ag, indicative of balanced charge transport for reducing hysteresis. The hole mobility (μ_h_) is derived from the equation J = (9/8)εε_0_μ_h_ (V^2^/L^3^) [54], and the plot ln(JL^3^/V^2^) versus electric field (V/L)^0.5^ is displayed in Appendix A. The calculated μ_h_ of the NiO_x_ thin film and nanoflakes are 2.58 × 10^−3^ and 3.42 × 10^−3^ cm^2^/Vs, respectively, implying that the PSC using the NiO_x_ nanoflakes HTL may have a higher J_SC_ value. The reduced charge recombination is also responsible for the augmented J_SC_, as described in the PL emission and TR-PL decay parts. Besides, the V_OC_ of the device using the NiO_x_ nanoflakes HTL is larger than that using the NiO_x_ thin film as the HTL, which is confirmed by the downshifted VB level. To verify the reproducibility of our PSCs, the statistical distribution of the four device parameters from 20 individual PSCs is depicted in Appendix A. All devices possessed good reproducibility, and those PSCs based on the NiO_x_ nanoflakes exhibited relatively higher photovoltaic performance. Figure 10c reveals the EQE spectra of devices using the NiO_x_ thin film or nanoflakes as the HTL and their integrated current density as a function of wavelength. The results show that the NiO_x_ nanoflakes-based device has higher photon-to-electron conversion efficiency from 300 to 800 nm than the NiO_x_ thin film. The integrated current density of devices based on the NiO_x_ thin film and nanoflakes was calculated to be 19.1 and 20.7 mA/cm^2^, respectively, which are close to their J_SC_ in Table 1. As mentioned in the Section 3.1, the transmittance of the NiO_x_ nanoflakes is somewhat lower than that of the NiO_x_ thin film. However, device performance is determined by many factors, not only the transmittance of charge transport layers. The SEM results confirm that perovskite layers with larger grains and fewer grain boundaries can be obtained on the NiO_x_ nanoflakes, reducing charge recombination in the boundaries. Therefore, the Jsc and EQE of devices based on the NiO_x_ nanoflakes are higher compared with the thin film. Although the best PCE of our device is moderate (~14%) compared with conventional silicon-based solar cells, our PSCs have several advantages, such as simple solution process, low manufacturing cost, and feasibility of large-area panels. Besides, less environmental pollution is produced during the manufacturing process of PSCs. In addition to the device efficiency, device stability of PSCs should also be considered as another important factor for real application. The unencapsulated PSCs were stored in a nitrogen glove box at 25 °C and measured in ambient air. After 60 days, the PSC using the NiO_x_ nanoflakes maintained 83% of its original efficiency, while the one based on the NiO_x_ thin film retained 75% of its initial PCE value, as shown in Figure 10d. The potential mechanism for the improved stability of the NiO_x_ nanoflakes devices can be explained as follows. The SEM result confirms that perovskite films with larger grains and fewer grain boundaries can be produced on the NiO_x_ nanoflakes. High-quality perovskite absorbers guarantee device performance and prevent short-term degradation of devices during storage. In 2020, Di Girolamo et al. reported the comparison of NiO nanoflakes by anodic electrodeposition and NiO films by sol-gel process [55]. They claimed that the morphology of the perovskite grown on top of electrodeposited NiO and sol-gel NiO was very similar. In this study, we proposed a different method to obtain NiO_x_ nanoflakes. We found that upper perovskites with larger grains and fewer boundaries were formed compared with the NiO_x_ thin film. Additionally, device stability tests were carried out, and the PSC based on the NiO_x_ nanoflakes showed better stability during storage. We conclude that the NiO_x_ nanoflakes can serve as a better HTL for PSCs with improved performance and stability.

## 4. Conclusions

We developed a three-stage method to prepare NiO_x_ nanoflakes as the efficient HTL for inverted PSCs. The nanostructured NiO_x_ exhibited high transmittance, good electrical conductivity, and matched energy level alignment, which led to effective hole extraction and transport. The perovskite film deposited on NiO_x_ nanoflakes had larger grain sizes and fewer boundaries, implying higher photogenerated current and fewer defects. Moreover, reduced PL intensity and shortened carrier lifetime reveal lower charge recombination, which is beneficial for photovoltaic application. The inverted device using NiO_x_ nanoflakes HTL showed a hysteresis-less behavior and a promising efficiency of 14.21%. After 60 days of storage, the PSC based on NiO_x_ nanoflakes HTL maintained 83% of its initial PCE value. Our results demonstrate the great potential of the NiO_x_ nanoflakes for the development of efficient and stable inverted PSCs.

## Figures and Tables

**Figure 1 nanomaterials-12-03336-f001:**
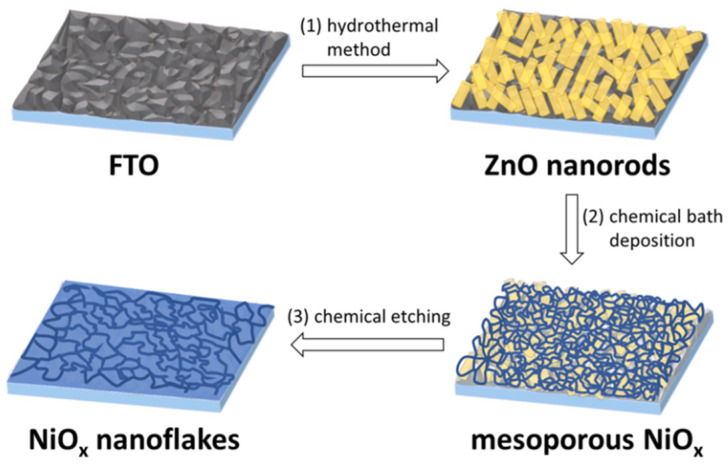
Schematic illustration of the preparation of NiO_x_ nanoflakes.

**Figure 2 nanomaterials-12-03336-f002:**
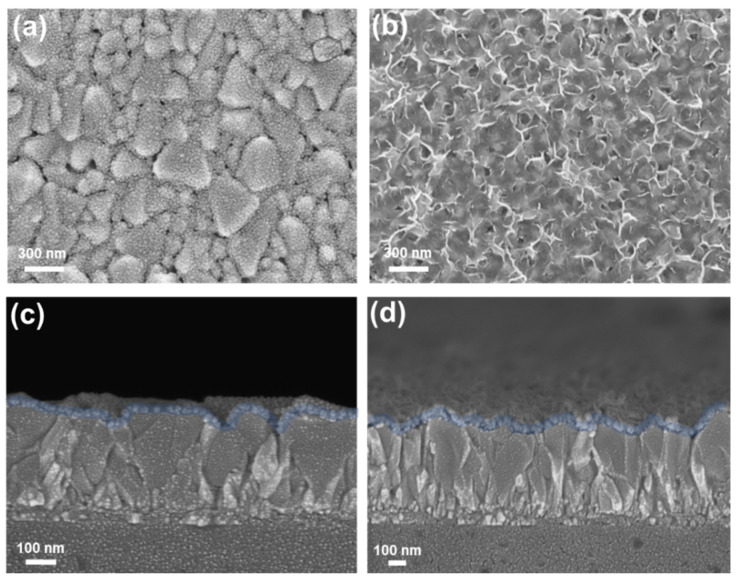
Top-view and cross-sectional SEM micrographs of the NiO_x_ (**a**,**c**) thin film and (**b**,**d**) nanoflakes deposited on FTO substrates.

**Figure 3 nanomaterials-12-03336-f003:**
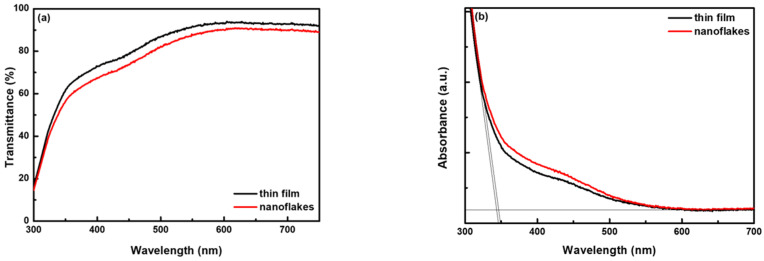
(**a**) Transmission and (**b**) absorption spectra of the NiO_x_ thin film and nanoflakes.

**Figure 4 nanomaterials-12-03336-f004:**
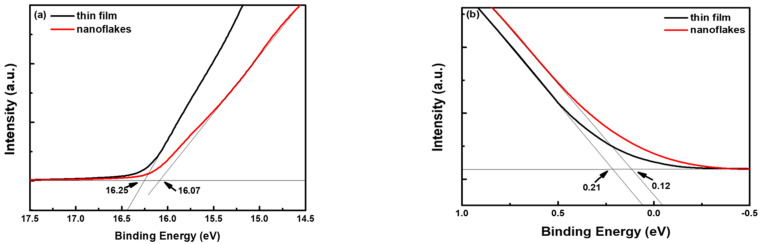
UPS spectra of the NiO_x_ thin film and nanoflakes in the (**a**) high and (**b**) low binding energy region.

**Figure 5 nanomaterials-12-03336-f005:**
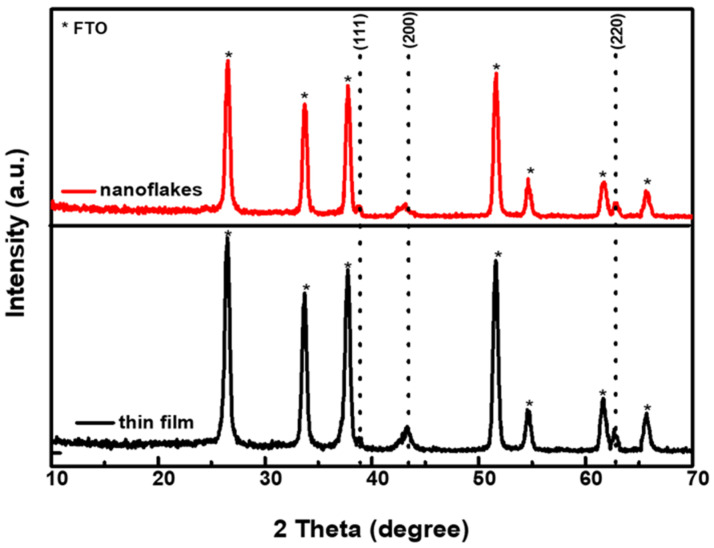
XRD patterns of the NiO_x_ thin film and nanoflakes on the FTO substrates.

**Figure 6 nanomaterials-12-03336-f006:**
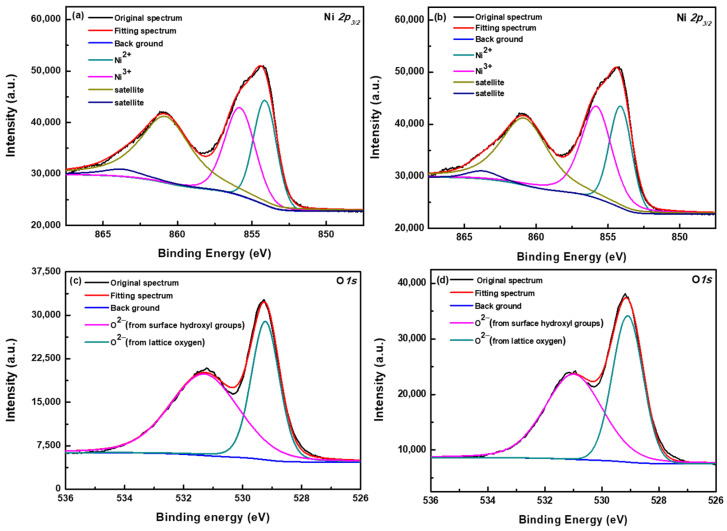
XPS spectra of Ni *2p_3/2_* and O *1s* elements in the (**a**,**c**) NiO_x_ thin film and (**b**,**d**) nanoflakes.

**Figure 7 nanomaterials-12-03336-f007:**
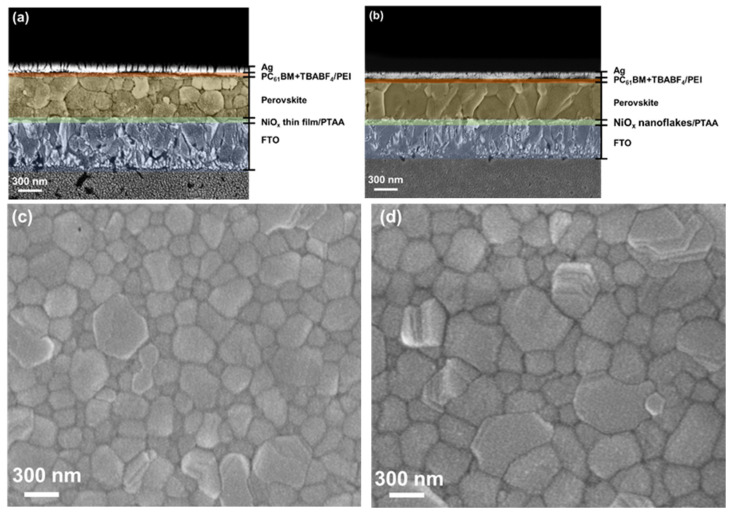
Cross-sectional SEM micrographs of PSCs based on the NiO_x_ (**a**) thin film and (**b**) nanoflakes as the HTL; top-view SEM micrographs of the perovskite deposited on the (**c**) NiO_x_ thin film/PTAA and (**d**) NiO_x_ nanoflakes/PTAA.

**Figure 8 nanomaterials-12-03336-f008:**
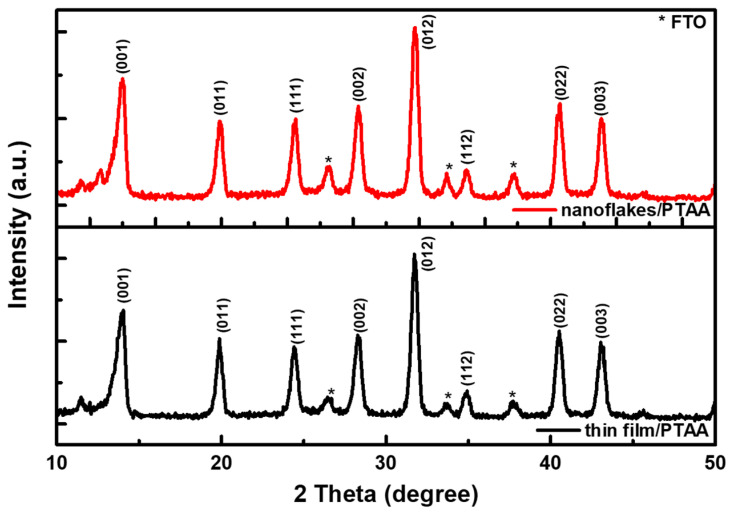
XRD patterns of the perovskite films on the NiO_x_ thin film/PTAA and nanoflakes/PTAA.

**Figure 9 nanomaterials-12-03336-f009:**
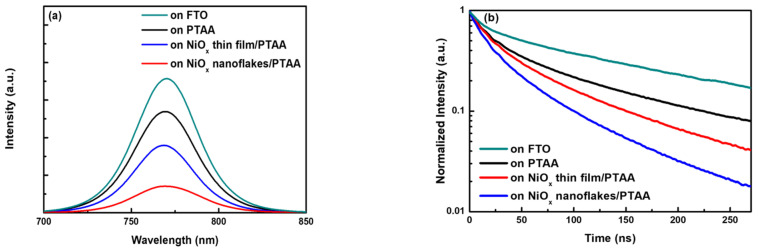
(**a**) PL emission spectra and (**b**) TR-PL decay curves of the perovskite on the FTO glass substrate, PTAA, NiO_x_ thin film/PTAA, and NiO_x_ nanoflakes/PTAA.

**Figure 10 nanomaterials-12-03336-f010:**
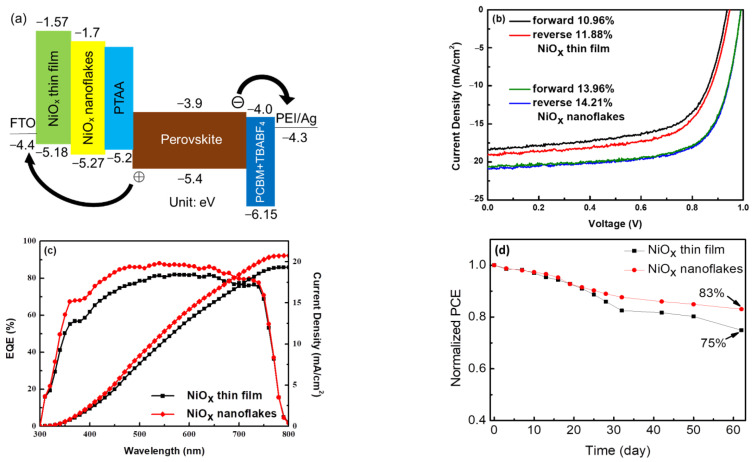
(**a**) Energy level diagram, (**b**) J–V characteristics, (**c**) EQE spectra and integrated current density, and (**d**) normalized PCE evolution of devices based on the NiO_x_ thin film and nanoflakes.

**Table 1 nanomaterials-12-03336-t001:** Device performance of inverted PSCs based on different NiO_x_ HTLs.

NiO_x_ HTL	Scan Direction	J_SC_ ^a^ (mA/cm^2^)	V_OC_ ^a^ (V)	FF ^a^ (%)	Best PCE ^a^ (%)	Avg PCE ^b^ (%)	R_S_ ^a^ (Ω·cm^2^)	R_Sh_ ^a^ (kΩ·cm^2^)
thin film	Forward	18.4	0.93	64	10.96	10.4	7.42	0.67
						(σ = 0.3)		
	Reverse	19.1	0.94	66	11.88	11.4	7.24	0.78
						(σ = 0.33)		
nanoflakes	Forward	20.5	0.99	68	13.96	13.7	5.85	1.36
						(σ = 0.15)		
	Reverse	20.8	0.99	69	14.21	13.9	5.67	1.53
						(σ = 0.17)		

^a^ The values were acquired from the champion device based on different HTLs. ^b^ Data were obtained from 20 devices. σ: standard deviation.

## Data Availability

Not applicable.

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
