# Peer review of "Preparation of Nickel Oxide Nanoflakes for Carrier Extraction and Transport in Perovskite Solar Cells"

_nanomaterials, 2022, doi:10.3390/nano12193336_

Round 1

Reviewer 1 Report

The paper of Chih-Yu Chang et.al is devoted to the material research of NiOx HTL layers, which can be of interest for photovoltaic applications. The best type of NiOx flakes based HTL was determined and photovoltaic properties of “NiOx-HTL” PSCs were measured.

Reviewer has the following remarks:

1) It is not clear from the text of the paper what could be the advantage of proposed “NiOx-HTL” PSCs over conventional silicon based solar cells. Measured value of photovoltaic efficiency (14%) is less than that in silicon solar cells. This is the major comment on the paper. The authors must clarify the prospects of their structures in photovoltaic applications.

2) The text contains typos, for example: Page 9 “which is accordance …” –> “which is in accordance …”

Author Response

(1) It is not clear from the text of the paper what could be the advantage of proposed “NiOx-HTL” PSCs over conventional silicon based solar cells. Measured value of photovoltaic efficiency (14%) is less than that in silicon solar cells. This is the major comment on the paper. The authors must clarify the prospects of their structures in photovoltaic applications.

Response: We thank for the reviewer’s comment and add more description to demonstrate the prospects of our PSCs. Although the best PCE of our device is moderate (~14%) as compared to conventional silicon-based solar cells, our PSCs have several advantages such as simple solution process, lower manufacturing cost, and feasibility of large-area panels. Besides, less environmental pollution is produced during the manufacturing process of PSCs. The above description has been added in the 3.3 section.

(2) The text contains typos, for example: Page 9 “which is accordance …” –> “which is in accordance …”

Response: Thanks for pointing this out. The word “in” has been inserted in the sentence. We also checked the whole text again to correct other typo/grammatical errors.

Reviewer 2 Report

In this study, Chang et al demonstrated a three-stage method to prepare NiOx nanoflakes for perovskite solar cells. The inverted perovskite solar cell devices using NiOx nanoflake as a HTL showed a hysteresis-less behavior and a promising efficiency in addition to the improved device stability. The results demonstrate great potential of the NiOx nanoflakes for the development of efficient and stable perovskite optoelectronics. This paper seems interesting for broad readership. I recommend this paper for publication after minor revision.

- It seems strange when comparing the results of Figure 10b and Table 1 (e.g. Voc). Also, in Table 1, the authors need to explain how each parameter was obtained (either from a champion device or an average of 20 devices).

- I recommend to include standard deviation in Figure 10d and Table 1.

 - Can authors give a short comment on the origin of improved device stability in the nanoflakes PSC?

Author Response

(1) It seems strange when comparing the results of Figure 10b and Table 1 (e.g. Voc). Also, in Table 1, the authors need to explain how each parameter was obtained (either from a champion device or an average of 20 devices).

Response: After re-checking the original data, we realized that the J-V characteristics of PSCs based on the NiOx nanoflakes in Figure 10b were incorrect. We apologize for this mistake and replace the old Figure 10b with a new one. The values of all device parameters in Table 1 are correct. The device parameters including Voc, Jsc, FF, best PCE, Rs, and Rsh were acquired from the champion device based on different HTLs. A new superscript symbol “a” and a short description were added in Table 1. The avg PCE values were obtained from 20 devices and the superscript symbol was modified to “b.”

(2) I recommend to include standard deviation in Figure 10d and Table 1.

Response: We thank for the reviewer’s suggestion and standard deviations for the average PCE has been added in Table 1. As for Figure 10d, the PCE values were recorded and plotted from the two champion devices using NiOx thin film and nanoflakes HTLs and therefore no standard deviation could be provided here.

(3) Can authors give a short comment on the origin of improved device stability in the nanoflakes PSC?

Response: The potential mechanism for the improved stability of the NiOx nanoflakes devices can be explained as follows. The SEM results confirm that perovskite films with larger grains and fewer grain boundaries can be produced on the NiOx nanoflakes. High-quality perovskite absorbers guarantee device performance and prevent short-term degradation of devices during storage. The above description has been added in the 3.3 section in the revised manuscript.

Reviewer 3 Report

The authors have presented a study on the development of NiOx nanoflakes layer for applying as an HTL on perovskite solar cells. The study contains a reasonable number of characterisations to back up the claims made in the manuscript. However, some areas need further clarifications and discussions to support the claims. The manuscript should be accepted if the following  comments are addressed properly.

1. In the second paragraph of Introduction, the authors have highlighted on the importance of the inverted p-i-n structure of perovskite solar cells. Lower hysteresis is indeed one of the advantages of p-i-n over n-i-p. However, the authors should also add a few sentences emphasising on the other advantages of the inverted structure such as: low temperature processability and compatibility with flexible substrates.

2. The overall English has further room for improvement. For example: Page 2 Line 83, “interlayer layer” repetition should be avoided. And Page 2 Line 76, “highly conductivity” should be replaced to “highly conductive”

3. How does the NiOx (thin films and nanoflakes) devices with and without PTAA differ. Further discussion should be added with accompanying characterisation results (AFM/SEM/XRD/J-V etc) perhaps with and without PTAA.

4. The cross-sectional SEM image (Figure 2 c and d) scalebar of 100 nm does not match with the thickness of the layers. The scalebar should be adjusted to become consistent with the claim of the measured thicknesses (30 nm and 58 nm).

5. The device area was not mentioned anywhere in the manuscript. It must be included in the J-V parameters discussion.

6. The transmittance of the NiOx nanoflakes were less compared to the thin films. However, the JSC and EQE for the NiOx nanoflakes containing devices were found to be higher compared with the thin films. Further discussion should be included to clarify why this is happening.

Author Response

(1) In the second paragraph of Introduction, the authors have highlighted on the importance of the inverted p-i-n structure of perovskite solar cells. Lower hysteresis is indeed one of the advantages of p-i-n over n-i-p. However, the authors should also add a few sentences emphasising on the other advantages of the inverted structure such as: low temperature processability and compatibility with flexible substrates.

Response: We thank for the reviewer’s comment and more description emphasizing the advantages of the p-i-n structure. The inverted structure offers the advantage of low temperature processability, thereby reducing manufacturing cost and particularly bringing compatibility with flexible substrates to expand diversity of photovoltaic cells. The above sentence has been added in the second paragraph of Introduction.

(2) The overall English has further room for improvement. For example: Page 2 Line 83, “interlayer layer” repetition should be avoided. And Page 2 Line 76, “highly conductivity” should be replaced to “highly conductive.”

Response: Thanks for pointing this out. The word “layer” in page 2, line 83 is deleted and “conductively” in page 2, line 76 is corrected with “conductive” in the revised manuscript.

(3) How does the NiOx (thin films and nanoflakes) devices with and without PTAA differ. Further discussion should be added with accompanying characterisation results (AFM/SEM/XRD/J-V etc) perhaps with and without PTAA.

Response: In the early stage of this study, we utilized sole NiOx thin film or nanoflakes as the HTL for fabricating PSCs and received lower device performance (PCE = 10.41% for NiOx thin film and 12.44% for NiOx nanoflakes). The J-V characteristics of devices without PTAA are provided in Figures S6(b) and (c). According to the previous literature [51], we realized that a thin layer of PTAA can be inserted between NiOx and perovskite layers to improve carrier transport ability and photovoltaic properties. Therefore, PTAA was incorporated in all devices and higher device performance was received. We believe that it is not necessary to collect other characterization results for those devices without PTAA. We should also explain that the apparatus like SEM or XRD belongs to the Precious Instruments Center at another university. All measurements must be reserved in advance and it takes more than one month to receive the data. We are afraid that there is not enough time to submit the revised version within this time frame. We regret that no AFM, SEM or XRD results of samples without PTAA can be provided here.

(4) The cross-sectional SEM image (Figure 2 c and d) scalebar of 100 nm does not match with the thickness of the layers. The scalebar should be adjusted to become consistent with the claim of the measured thicknesses (30 nm and 58 nm).

Response: Thanks for pointing this out. The scale bar in Figures 2(c) and (d) has been adjusted in the revised manuscript.

(5) The device area was not mentioned anywhere in the manuscript. It must be included in the J-V parameters discussion.

Response: Actually, the effective area of each device (4.5 mm2) is indicated in the 2.3 Device fabrication section. For clarity, the device area is mentioned again in the J-V parameters discussion in the 3.3 section.

(6) The transmittance of the NiOx nanoflakes were less compared to the thin films. However, the JSC and EQE for the NiOx nanoflakes containing devices were found to be higher compared with the thin films. Further discussion should be included to clarify why this is happening.

Response: We thank for the reviewer’s comment. As mentioned in the 3.1 section, the transmittance of the NiOx nanoflakes is somewhat lower than that of the NiOx thin film. However, device performance is determined by many factors, not only the transmittance of charge transport layers. The SEM results confirm that perovskite layers with larger grains and fewer grain boundaries can be obtained on the NiOx nanoflakes, reducing charge recombination in the boundaries. Therefore, the Jsc and EQE of devices based on the NiOx nanoflakes are higher compared with the thin film. The above description has been added in the 3.3 section in the revised manuscript.

Reviewer 4 Report

The authors have developed NiOx nanoflakes as the hole transport layer (HTL) in perovskite solar cells. NiOx nanoflakes have been fabricated by a three-stage method starting from Ni precursors on ZnO nanorods. NiOx nanoflakes performances were evaluated in inverted perovskites solar cells, and authors demonstrated accurately that NiOx nanoflakes outperforms thin films NiOx as HTL.

The manuscript is clear, well written and the characterization of NiOx materials and the perovskite solar cells is complete.

 A few minor suggestions to improve manuscript quality are given below.

1) Please describe in more details how the NiOx thin films (not the NiOx nanoflakes) were fabricated.

2) Regarding the ratio of Ni3+/Ni2+ of thin films and nanoflakes, please explain better the sentence at lines 271-272. If the Ni3+/Ni2+ is higher in NiOx nanoflakes than in thin films it means that a higher number of holes is present in the nanoflakes. Why the authors refer to transport ability of NiOx?

3) I suggest the reading of the following manuscript “Anodically electrodeposited NiO nanoflakes as hole selective contact in efficient air processed p-i-n perovskite solar cells”, Solar Energy Materials and Solar Cells 205:110288, 2019,  where the authors report about the comparison of NiO nanoflakes and sol- gel NiO films.

Author Response

(1) Please describe in more details how the NiOx thin films (not the NiOx nanoflakes) were fabricated.

Response: We thank for the reviewer’s comment. To prepare NiOx thin films, a precursor solution consisting of Ni(OAc)2•4H2O (0.124 mg) and ethanolamine (30 μL) in 5 mL of isopropyl alcohol (IPA) was heated at 70 °C with stirring in a sealed glass vial overnight. The precursor solution was then spin-coated on the FTO substrate, followed by calcination at 450 °C to obtain NiOx thin films. The above description about the preparation of NiOx thin films are added in the 2.2 section. The title of the 2.2 section is also changed to “Preparation of NiOx thin films and nanoflakes.”

(2) Regarding the ratio of Ni3+/Ni2+ of thin films and nanoflakes, please explain better the sentence at lines 271-272. If the Ni3+/Ni2+ is higher in NiOx nanoflakes than in thin films it means that a higher number of holes is present in the nanoflakes. Why the authors refer to transport ability of NiOx?

Response: We thank for the reviewer’s comment. It is certain that higher Ni3+/Ni2+ ratio means a higher number of holes in the NiOx nanoflakes than in thin film. The reason for the higher hole transport ability in the NiOx nanoflakes is described as follows. The higher Ni3+ concentration in the NiOx nanoflakes implies the existence of interstitial oxygen and Ni vacancies in the crystalline structure. Compared to the Ni2+, the higher valence Ni3+ means a lack of one electron that produces a hole carrier. These hole carriers can move through the crystal by hopping from the Ni3+ site to another one or by moving in a narrow polaron band, indicative of enhanced hole transport ability. The above description has been added in the XPS discussion in the 3.1 section.

(3) I suggest the reading of the following manuscript “Anodically electrodeposited NiO nanoflakes as hole selective contact in efficient air processed p-i-n perovskite solar cells”, Solar Energy Materials and Solar Cells 205:110288, 2019,  where the authors report about the comparison of NiO nanoflakes and sol-gel NiO films.

Response: We thank for the reviewer’s suggestion. The suggested article is added as ref [55] and a short discussion is provided in the 3.3 section. In 2020, Di Girolamo et al. reported the comparison of NiO nanoflakes by anodic electrodeposition and NiO films by sol-gel process. They claimed that the morphology of the perovskite grown on top of electrodeposited NiO and sol-gel NiO was very similar. In this study, we proposed a different method to obtain NiOx nanoflakes and found that upper perovskites with larger grains and fewer boundaries were formed compared to the NiOx thin film. Additionally, device stability tests were carried out and the PSC based on the NiOx nanoflakes showed better stability during storage. The above description has been added in the last paragraph in the 3.3 section.